# Cuproptosis-Related MiR-21-5p/FDX1 Axis in Clear Cell Renal Cell Carcinoma and Its Potential Impact on Tumor Microenvironment

**DOI:** 10.3390/cells12010173

**Published:** 2022-12-31

**Authors:** Mingyue Xie, Bo Cheng, Shuang Yu, Yajie He, Yu Cao, Tiejun Zhou, Kun Han, Rongyang Dai, Ronghao Wang

**Affiliations:** 1Department of Biochemistry and Molecular Biology, School of Basic Medical Sciences, Southwest Medical University, Luzhou 646000, China; 2Department of Urology, The Affiliated Hospital of Southwest Medical University, Luzhou 646000, China; 3Sichuan Clinical Research Center for Nephropathy, The Affiliated Hospital of Southwest Medical University, Luzhou 646000, China; 4Department of Pathology, The Affiliated Hospital of Southwest Medical University, Luzhou 646000, China

**Keywords:** renal cell carcinoma, cuproptosis, FDX1, miR-21-5p, microenvironment

## Abstract

As a newly identified type of programmed cell death, cuproptosis may have an impact on cancer development, including clear cell renal cell carcinoma (ccRCC). Herein, we first noticed that the expression levels of cuproptosis regulators exhibited a tight correlation with the clinicopathological characteristics of ccRCC. The cuproptosis-sensitive sub-type (CSS), classified via consensus clustering analysis, harbored a higher overall survival rate compared to the cuproptosis-resistant sub-type (CRS), which may have resulted from the differential infiltration of immune cells. FDX1, the cuproptosis master regulator, was experimentally determined as a tumor suppressor in ccRCC cells by suppressing the cell growth and cell invasion of ACHN and OSRC-2 cells in a cuproptosis-dependent and -independent manner. The results from IHC staining also demonstrated that FDX1 expression was negatively correlated with ccRCC tumor initiation and progression. Furthermore, we identified the miR-21-5p/FDX1 axis in ccRCC and experimentally verified that miR-21-5p directly binds the 3′-UTR of FDX1 to mediate its degradation. Consequently, a miR-21-5p inhibitor suppressed the cell growth and cell invasion of ACHN and OSRC-2 cells, which could be compensated by FDX1 knockdown, reinforcing the functional linkage between miR-21-5p and FDX1 in ccRCC. Finally, we evaluated the ccRCC tumor microenvironment under the miR-21-5p/FDX1 axis and noted that this axis was strongly associated with the infiltration of immune cells such as CD4^+^ T cells, Treg cells, and macrophages, suggesting that this signaling axis may alter microenvironmental components to drive ccRCC progression. Overall, this study constructed the miR-21-5p/FDX1 axis in ccRCC and analyzed its potential impact on the tumor microenvironment, providing valuable insights to improve current ccRCC management.

## 1. Introduction

Cell death is precisely regulated at the molecular level within the human body. Apoptosis, necroptosis, pyroptosis, and ferroptosis are well-defined ways by which cells undergo death in certain circumstances [1,2,3,4]. Recently, a novel type of cell death named cuproptosis was functionally identified [5] by Todd Golub’s group. Distinct from other types of cell death, cuproptosis refers to copper-ionophore-mediated cell death and cannot be abrogated by any current cell death inhibitors [5]. According to the literature, cuproptosis is tightly controlled by the copper ion concentration and protein lipoylation: the direct binding of excessive copper ions to the lipoylated proteins will cause the aggregation of lipoylated proteins and lead to the loss of iron–sulfur cluster proteins, resulting in proteotoxic stress and triggering cell death. In addition, 10 genes have been identified as being closely associated with cuproptosis. Among them, FDX1, LIAS, LIPT1, DLD, DLAT, PDHA1, and PDHB positively promote cuproptosis, while MTF1, GLS, and CDKN2A play negative roles in the process of cuproptosis.

As the master regulator of cuproptosis, FDX1, which was originally documented as an electron transfer to regulate the biosynthesis of steroid hormones, vitamin D, and bile acid in urological tissues, the kidney and the liver, respectively [6], plays a central role in connecting copper toxicity with protein lipolyation, two essential steps indispensable for cuproptosis initiation. On the one hand, FDX1 encodes a reductase to reduce Cu^2+^ to its toxic form Cu^1+^. On the other hand, FDX1 acts as the upstream regulator of the lipoic acid pathway to control protein lipolyation. Finally, toxic copper binds to lipolyated proteins such as DLAT and promotes their oligomerization, triggering cuproptosis. Recent studies also highlight the involvement of FDX1, as well as its associated cuproptosis, in the development of various cancers. For example, FDX1 or the cuproptosis signature serves as a prognostic indicator of various cancers, including bladder cancer [7], hepatocarcinoma [8], melanoma [9], and breast cancer [10], suggesting that its participation in cancer progression deserves a close investigation.

Clear cell renal cell carcinoma (ccRCC) accounts for an estimated 80 percent of kidney cancers and is mainly caused by inactivation of Von Hippel–Lindau (VHL) [11]. As a tumor suppressor, VHL functions as an E3 ubiquitin ligase to mediate the proteasomal degradation of hypoxia-inducible factor (HIF) [12,13]. In ccRCC, the inactivation of VHL leads to the protein accumulation of HIF-1a and HIF-2a, which translocate to the nucleus and regulate a variety of gene expression programs at the transcriptional level to promote tumor growth, tumor metastasis, and neo-angiogenesis [14,15]. This deregulation led to the identification of tyrosine kinase inhibitors, such as sunitinib and pazopanib, to treat ccRCC patients [16,17]. However, inherited and acquired resistance to cell death minimizes the anti-cancer effects of therapeutic drugs and receives broad attention. Whether ccRCC harbors a resistance to cuproptosis, and the role of FDX1-dependent or -independent of cuproptosis in ccRCC development have not been investigated yet.

In this study, we aimed to examine the clinical values of cuproptosis regulators and investigate the contributing role of FDX1 in ccRCC. Our data showed that cuproptosis-sensitive patients experienced longer survival as compared with their cuproptosis-resistant counterparts, which may be a result of the considerable infiltration of CD4^+^ T cells. As the master regulator of cuproptosis, FDX1 was experimentally determined as a tumor suppressor to inhibit the cell growth and invasion of ccRCC cells. FDX1 expression was also highly correlated with the T stage, tumor grade, and CD4^+^ T cell infiltration in the collected ccRCC samples. In addition, the fact that FDX1 was post-transcriptionally regulated according to our analyses prompted us to computationally establish a linkage between miR-21-5p and FDX1, which was validated by a direct-binding-mediated FDX1 reduction. Importantly, a tight correlation between immune cell infiltration and the expression levels of miR-21-5p and FDX1 was evidently observed in ccRCC, indicating that this signaling may potently affect the ccRCC microenvironment. Together, all these findings highlight that resistance to cuproptosis under the miR-21-5p/FDX1 axis, as well as its educated tumor microenvironment is a potential mechanism driving ccRCC progression, and recovery of cuproptosis may be an alternative strategy to overcome this type of disease.

## 2. Materials and Methods

### 2.1. Data Collection

RNA sequencing datasets and their related clinical information were downloaded from The Cancer Genome Atlas (TCGA, https://portal.gdc.cancer.gov (accessed on 20 March 2022)) and Gene Expression Omnibus (GEO, http://www.ncbi.nlm.nih.gov/geo/ (accessed on 16 June 2022)). The DNA Methylation dataset (KIRC, Illumina Human 450 (Illumina Inc., San Diego, CA, USA)) was obtained from UCSC Xena (http://xena.ucsc.edu/ (accessed on 10 April 2022)).

### 2.2. Bioinformatics Analyses

The “survival” R package was used to assess the overall survival of ccRCC in TCGA-KIRC. The “tinyarray” R package was applied to divide TCGA samples into the “Tumor” or “Normal” group. The “ConsensusClusterPlus” R package was utilized to classify TCGA KIRC into two clusters using cuproptosis regulators as baits, and then the data were visualized in two dimensions via principal component analysis (PCA) and t-distributed stochastic neighbor embedding (t-SNE) analysis with the “ggplot2” and “Rtsne” packages, respectively. The expression pattern of cuproptosis regulators in two different clusters was also visualized using the “pheatmap” R package.

For cuproptosis risk score generation, the coefficients of FDX1, LIAS, DLD, DLAT, PDHB, MTF1, and CDKN2A were gained from multivariate Cox regression analysis of TCGA-KIRC, and the cuproptosis risk score was calculated using the following equation: risk score = ∑ gene expression X coefficient. TCGA-KIRC ccRCC patients were divided into high-score and low-score groups according to the optimal cutoff point calculated using the “survminer” R package.

For functional enrichment analyses of CSS and CRS, Gene Ontology (GO) and the Kyoto Encyclopedia of Genes and Genomes (KEGG) were used to determine the biological functions related to cuproptosis sensitivity using the “stringr”, “enrichplot”, “clusterProfiler”, “GOplot”, “DOSE”, “ggnewscale”, and “topGO” packages. Gene set enrichment analysis (GSEA) was performed using GSEA software (UC San Diego and Broad Institute, Cambridge, MA, USA).

For DNA methylation analysis, the DNA methylation data (TCGA.KIRC.sampleMap/HumanMethylation450) were first normalized using the “minfi” package, and then the methylation levels on the promoter regions of cuproptosis regulators including FDX1, DLD, DLAT, PDHB, LIAS, LIPT1, MTF1, GLS, PDHA1, and CDKN2A were visualized using “ggpubr”.

For immune cell infiltration analysis, the LM22 signature matrix and CIBERSORT R script were downloaded from the previous document [18], and the samples from CSS and CRS were used to evaluate the level of immune cell infiltration based on the signature matrix related to the specific population of immune cells and the CIBERSORT algorithm.

### 2.3. Cell Culture

OSRC-2 (Cat: IM-H061) and ACHN (Cat: IM-H058), representing VHL null and VHL wild-type RCC cell lines, respectively, were purchased from the Immocell Biotechnology company (Xiamen, China). Cells were maintained in 10% FBS DMEM supplemented with 2 mM L-glutamine, 100 IU/mL penicillin, and 100 µg/mL streptomycin. Cells were cultured in a 37 °C/5% CO_2_ humidified hood.

### 2.4. siRNA Transfection

SiRNAs were purchased from General Biology Company (Anhui, China) and transfected into cells using lipofectamine 3000 as a reagent (Invitrogen, Waltham, MA, USA) at a 50 nM concentration. siRNA sequences against FDX1 are listed as follows:

siFDX1#1 sense, 5′-CAUUAACAACCAAAGGAAATT-3′;

siFDX1#1 antisense, 5′-UUUCCUUUGGUUGUUAAUGTT-3′;

siFDX1#2 sense, 5′-GCCAAAUCUGUUUGACAAATT-3′;

siFDX1#2 antisense, 5′-UUUGUCAAACAGAUUUGGCTT-3′;

siFDX1#3 sense, 5′-GAUAGAAAAACCUUACAUATT-3′;

siFDX1#3 antisense, 5′-UAUGUAAGGUUUUUCUAUCTT-3′.

### 2.5. Western Blotting

Cells with or without siFDX1 were lysed with a RIPA buffer (30 mM HEPES, pH 7.4, 150 mM NaCl, 1% Nonidet P-40, 0.5% sodium deoxycholate, 0.1% SDS, 5 mM EDTA, 1 mM NaVO4, 50 mM NaF, 1 mM PMSF, 10% pepstatin A, 10 µg/mL leupeptin, and 10 µg/mL aprotinin). An amount of 20 µg of protein was separated by 15% SDS-PAGE and transferred onto a nylon membrane. The membranes were blocked in 5% milk TBST for 30 min at room temperature, blotted using the FDX1 antibody (1:1000, 12592-1-AP, Proteintech, Rosemont, IL, USA) overnight at 4 °C, followed by conjugated secondary antibodies, and visualized using ODYSSEY CLX (LI-COR, Lincoln, NE, USA).

### 2.6. Real-Time Quantitative PCR (RT-qPCR)

Total RNA was extracted using TRIzol reagent (TIANGEN BIOTECH Co., Ltd., Beijing, China), and 1 µg of RNA was subjected to reverse transcription with the ReverTra Ace™ qPCR RT Kit (TOYOBO, Osaka, Japan). Real-time quantitative PCR was performed in a LightCycler 480 with QuantiNova SYBR Green dye (TOYOBO, Osaka, Japan). The primers used in this study are listed as follows:

FDX1 forward, 5′-TTCAACCTGTCACCTCATCTTTG-3′;

FDX1 reverse, 5′-TGCCAGATCGAGCATGTCATT-3′.

### 2.7. CCK8 Assay

RCC cells with or without siFDX1 were seeded into 96-well plates at a 5000 cells/well concentration. Cell viability was assessed using a Cell-Counting Kit-8 (CCK8) assay (APE*BIO, Lot NO.K101823133EF5E) at 0, 1, 2, and 3 days.

### 2.8. Transwell Invasion Assay

Standard Matrigel (356235, Corning) was diluted with serum-free DMEM and seeded in an 8 µm-pore-sized upper chamber (Corning, Inc., Corning, NY, USA). RCC cells with or without siFDX1 were then harvested and seeded into the upper chambers at a 5 × 10^4^ cells/well concentration, and 10% FBS DMEM medium was added to the bottom chambers as an attractant. After 16 h, the invaded cells were fixed with 75% ethanol and stained with 0.1% crystal violet. The number of invaded cells was determined using image J software 1.53v downloaded from the National Institutes of Health (NIH, Bethesda, MD, USA).

### 2.9. Copper Ion Detection

The Cu^2+^ level within 1 × 10^4^ OSRC-2 cells w/wo FDX1 or miR-21-5p was detected using a Cu colorimetric assay kit (E-BC-K300-M, Elabscience, Wuhan, China). A standard curve was plotted, and the Cu^2+^ level was determined based on the absorbance value at 580 nm.

### 2.10. Immunohistochemical Staining (IHC)

Deparaffinized ccRCC microarray sections (*n* = 62, the clinicopathological characteristics were listed in Appendix A) were incubated with 3% peroxidase methanol for 15 min at room temperature, followed by antigen retrieval in citrate buffer (pH = 6.0) for 10 min in a microwave. Sections were then blocked with 5% BSA + 5% milk PBS for 1 h at room temperature, followed by incubation with anti-FDX1 (1:100, 12592-1-AP, Proteintech, Rosemont, IL, USA) or anti-CD4 (25229, CST, Danvers, MA, USA) antibodies at 4 °C overnight. The DAB kit (Solarbio, Beijing, China) was utilized to determine FDX1 and CD4 signals. The FDX1 IHC score was calculated using ImageJ software. Patient samples were collected, and experiments were performed with the approval of the Ethics Committee of the Affiliated Hospital of Southwest Medical University (ID: KY2022337).

### 2.11. Statistics

Statistical analyses included a one-way ANOVA test using Graphpad Prism 8.0.2 (San Diego, CA, USA). A chi-square test was used to analyze the significant differences in FDX1 IHC scores in different histological T stages or grades. A log-rank test was utilized to determine the statistical significance of survival between different groups. Linear correlation analysis was performed to assess the correlation between the FDX1 IHC score and the CD4^+^ T cell population. A value of *p* < 0.05 was considered statistically significant.

## 3. Results

### 3.1. Clinical Significance of Cuproptosis Regulators in ccRCC

To pursue the clinical values of cuproptosis regulators in ccRCC, we first examined their expression levels in ccRCC patients of TCGA-KIRC compared to normal kidney counterparts. As shown in Figure 1A, the expression levels of 5/7 positive regulators of cuproptosis (FDX1, DLD, DLAT, PDHB, and PDHA1) were robustly decreased in ccRCC as compared to normal kidney controls, whereas the negative regulator CDKN2A was profoundly increased in ccRCC patients as compared to the adjacent non-cancerous tissues, suggesting that the carcinogenesis of ccRCC selectively suppresses physiological cuproptosis signaling. Further investigation demonstrated that the differential expression level of individual genes between ccRCC and normal kidney tissues could not be explained by the differential DNA methylation level (Figure 1B), suggesting that post-transcriptional regulations may be involved. Significantly, the high expression levels of these seven positive regulators of cuproptosis served as good indicators of the overall survival of ccRCC patients based on Kaplan–Meier survival analyses (Figure 1C–I), indicating that resistance to cuproptosis is one potential mechanism driving ccRCC progression.

We therefore conducted a complete evaluation of the associations between the expression levels of cuproptosis regulators and the clinicopathological features of ccRCC using the TCGA-KIRC dataset. As shown in Appendix A, positive regulators of cuproptosis, including FDX1, LIAS, DLD, DLAT, and PDHB, were decreasingly expressed as ccRCC progressed to an advanced histological T stage. In contrast, the negative regulators MTF1 and CDKN2A were examined and found to be highly expressed in the later histological T stage of ccRCC. A significantly negative correlation of LIAS and DLAT with ccRCC grade and a positive correlation of CDKN2A with ccRCC grade were also clearly observed. In addition, the expression levels of FDX1, LIAS, and DIAT were remarkably decreased in ccRCC patients with distant metastasis. Similarly, expression levels of FDX1, DLD, DIAT, and PDHB were much more abundant in alive ccRCC patients than those in dead controls, while the expression levels of CDKN2A and MTF1 were oppositely presented in this profile. Of note, no evident correlations were found between cuproptosis regulators and age, gender difference, and lymph node metastasis.

Together, the data in Figure 1A–I support the notion that cuproptosis regulators are highly correlated with the clinicopathological parameters of ccRCC, and that mediated physiological cuproptosis may play a certain role in ccRCC development.

### 3.2. Consensus Clustering Analysis to Evaluate the Potential Role of Cuproptosis in ccRCC

To explore the potential role of physiological cuproptosis in ccRCC development, we classified TCGA-KIRC ccRCC patients by conducting consensus clustering analysis using cuproptosis regulators as baits. According to the CDF curve and delta area, k = 2 was chosen as the category number to classify ccRCC patients, which were clearly distinguished based on the PCA evaluation (Figure 2A–C). A heatmap of the expression patterns of cuproptosis regulators in cluster 1 and cluster 2 is displayed in Figure 2D, which indicates that 7 positive cuproptosis regulators were highly enriched in cluster 1, while CKDN2A was enriched in cluster 2. For a convenient description, we refer to cluster 1 as the cuproptosis-sensitive sub-type (CSS) and cluster 2 as the cuproptosis-resistant sub-type (CRS). We then compared the clinicopathological characteristics of CSS and CRS. The data illustrated that CRS ccRCC patients were prone to the risk of having worse clinicopathological features including the T stage, grade, and metastatic and alive status (Figure 2E). More importantly, CRS ccRCC patients had a shorter overall survival compared to CSS cohorts (Figure 2F, *p* = 0.00026), implying that physiological cuproptosis benefits patients’ life expectancy.

To examine which signaling pathways were associated with cuproptosis sensitivity in ccRCC, we performed GO and KEGG analyses of CSS and CRS. As a result, pathways such as epidermis development, skin development, humoral immune response, etc., in BP (Biological Process), collagen-containing extracellular matrix in CC (Cellular Component) and receptor ligand activity, signaling receptor activator activity, etc., in MF (Molecular Function) were significantly enriched to be correlated with cuproptosis sensitivity according to the GO analysis (Appendix A). Neuroactive ligand receptor interaction, estrogen signaling pathway, staphylococcus aureus infection, synaptic vesicle cycle, IL-17 signaling pathway, and collecting duct acid secretion were determined by KEGG pathway analysis to be potentially associated with cuproptosis sensitivity in ccRCC (Appendix A).

Next, we conducted an immune cell infiltration analysis of CSS and CRS to establish a relationship between physiological cuproptosis and the tumor microenvironment. We first noticed that T cells and macrophages were the most abundant immune cells surrounding ccRCC (Figure 2G,H), in line with their potent roles in regulating the immune response. Among T cell populations, T regulatory Treg cells, considered as immunosuppressive cells, were found to be less infiltrated in CSS ccRCC than in CRS control (Figure 2I), suggesting preferable infiltration of T regulatory Treg cells into cuproptosis-resistant ccRCC to support tumor growth. In contrast, CD4^+^ T memory resting cells in CSS were much more abundant than those in CRS (Figure 2I). Previous studies have reported that CD4^+^ T memory resting cells may be activated upon external stimuli and exert anti-cancer activity via activating CD8^+^ T cytotoxic cells [19]. Activated CD4^+^ T cells could also mediate cytotoxicity against tumors, similar to their counterparts. Hence, ccRCC patients sensitive to cuproptosis may have much more potential to initiate a T cell immune response upon external stimuli than patients resistant to cuproptosis.

### 3.3. Cuproptosis Risk Score Has a Prognostic Value in ccRCC

To generate a prognostic model to predict ccRCC overall survival, we first applied a univariate Cox regression analysis towards 10 cuproptosis regulators in TCGA-KIRC ccRCC patients and found that seven members (FDX1, LIAS, DLD, DLAT, PDHB, MTF1, and CDKN2A) were significantly correlated with the overall survival of ccRCC patients (Appendix A), which were then subjected to multivariate Cox regression analysis (Figure 3A). As a result, a cuproptosis risk score was generated for each individual ccRCC patient by weighing the prognostic input of individual genes based on their Cox regression coefficients and expression levels. Subsequently, ccRCC patients were classified into two groups: cuproptosis_score high and cuproptosis_score low, using a risk score = 3.52 as a cutoff point based on the analysis from the survcutpoint in the survminer R package (Appendix A). A Kaplan–Meier survival analysis of these two groups demonstrated that cuproptosis_score high patients harbored a higher overall survival rate than their cuproptosis_score low counterparts (Figure 3B), the fitness of which was confirmed by an overall survival analysis of an independent ccRCC dataset (Figure 3C, GEO29609) as well as AUC analysis (Figure 3D, AUC = 0.684). Consistent with the above results, the cuproptosis risk score was positively correlated with the CD^4+^ T memory resting cell population and negatively correlated with the T regulatory Treg cell population (Figure 3E,F). Together, these data suggest that the cuproptosis risk score can be utilized as a prognostic factor to predict ccRCC overall survival.

### 3.4. Experimental Validation Supports the Tumor-Suppressive Role of FDX1 in ccRCC

The cuproptosis-dependent and -independent roles of FDX1 in ccRCC development have not been explored yet. To this end, we first examined FDX1 expression in a collection of 10 paired ccRCC masses, which showed that FDX1 was remarkably reduced in ccRCC masses as compared with the adjacent non-cancerous controls (Figure 4A). In line with this, IHC staining of ccRCC tissue micro-array (27/62 cRCC with paired adjacent kidney tissues) also confirmed the decreased expression of FDX1 in ccRCC (Figure 4B, *n* = 27), suggesting that FDX1 may serve as a tumor-suppressive regulator in ccRCC development. To test whether FDX1 manipulation could alter the biological activities of ccRCC cells, we utilized three siRNAs to knock down FDX1 in OSRC-2 cells, which efficiently silenced FDX1 at both the mRNA (Figure 4C) and protein levels (Figure 4D). As shown in Figure 4E, an increased cell viability was observed in FDX1-depleted OSRC-2 and ACHN cells. Moreover, the FDX1 reduction also profoundly increased the cell invasion of OSRC-2 and ACHN cells (Figure 4F). All these data suggest that FDX1 decreased the cell growth and cell invasion of ccRCC cells, dependent on or independent of cuproptosis.

To evaluate the cuproptosis-independent role of FDX1, we performed a GSEA analysis of TCGA-KIRC and found that the JAK_STAT pathway was significantly enriched in FDX1_low ccRCC patients. To confirm this, we knocked down FDX1 and examined JAK_STAT signaling by detecting the p-STAT3 (Y705) level. As shown in Figure 4G, the phosphorylation level of STAT3 at Y705 was clearly boosted in FDX1-depleted OSRC-2 cells, implying that siFDX1-mediated cell growth and cell invasion may be partially attributable to the activation of STAT3 signaling.

Taken together, these experimental data suggest that FDX1 functions as a tumor suppressor to influence ccRCC development at least by silencing physiological cuproptosis and triggering STAT3 signaling.

### 3.5. FDX1 Correlates with the Infiltration of CD4^+^ T Cells and Prognosis in ccRCC

To pursue the clinical value of FDX1 in ccRCC, we first correlated the FDX1 IHC intensity with the histological T stage and tumor grade of ccRCC tumors in the micro-array (*n* = 51, we excluded 11 tumors because their T stage and grade were not confidently confirmed by a pathologist). As shown in Figure 5A–C, FDX1 was preferably expressed at a higher level in ccRCC patients with either histological T stage < T2 or grade G1 + G2, indicating that FDX1 is a good prognostic factor in ccRCC progression. Previous bioinformatic analysis illuminated that cuproptosis-sensitive ccRCC patients were considerably infiltrated with CD4^+^ T memory resting cells, which prompted us to investigate the correlation between FDX1 and the CD4^+^ T cell population in ccRCC samples. To achieve this, we stained the CD4^+^ T cell population in the ccRCC tissue microarray using an anti-CD4 antibody (*n* = 62) and correlated it with the FDX1 IHC score. The results revealed that there was a positive correlation between the FDX1 IHC score and the CD4^+^ T cell population (Figure 5D,E), strengthening the notion that the CD4^+^ T cell immune response may be strongly responsive to FDX1-mediated cuproptosis in ccRCC. In agreement with this result, analysis of the TCGA KIRC dataset also confirmed that FDX1 was positively correlated with the CD4^+^ T cell population (Figure 5F, left). Of note, a positive correlation between FDX1 expression and the CD8^+^ T cell population was also observed (Figure 5F, Right). Collectively, these data imply that FDX1 expression is positively associated with the CD4^+^ T cell population and that T cell immune responses may be implicated in FDX1-mediated ccRCC tumor suppression.

### 3.6. Identification of miR-21-5p as the Upstream Regulator of FDX1

The aforementioned analyses suggested that the reduction in FDX1 may be caused by post-transcriptional regulation. Since miRNA-mediated mRNA degradation can be a critical post-transcriptional event, we sought to identify which miRNAs could potentially target FDX1 mRNA. To this end, we first searched for miRNAs that were differentially expressed between normal and ccRCC tissues in TCGA-KIRC (Figure 6A) as our candidates. Next, the prognosis-based LASSO algorithm identified 14 miRNAs that could potentially affect ccRCC progression (Figure 6B,C). MiR-21-5p was finally screened as one promising candidate to target FDX1 mRNA based on the miRDB prediction software (Figure 6D), and its expression was negatively correlated with the overall survival time in ccRCC (Figure 6E). To validate whether miR-21-5p could indeed target FDX1 mRNA, we introduced a miR-21-5p inhibitor into ACHN and OSRC-2 cells and observed a robust increase in FDX1 in these cells compared to NC-treated corresponding controls (Figure 6F). Furthermore, a 3′-UTR-based luciferase reporter assay demonstrated that miR-21-5p mimics remarkably suppressed the activity of luciferase with the wild-type 3′-UTR of FDX1 mRNA, which was abrogated when the miR-21-5p binding site was deleted (Figure 6G). We also observed that the miR-21-5p inhibitor profoundly slowed down the cell growth and cell invasion of ACHN and OSRC-2 cells, which could be rescued by FDX1 siRNA (Figure 6H,I). Importantly, FDX1 knockdown in OSRC-2 cells led to Cu^2+^ accumulation, whereas OSRC-2 cells exposed to the miR-21-5p inhibitor had a lower Cu^2+^ level than the control cells (Figure 6J), suggesting that the miR-21-5p-mediated ccRCC phenotype at least partially depends on FDX1-mediated cuproptosis. Indeed, miR-21-5p was inversely correlated with FDX1 in the TCGA-KIRC dataset (Figure 6K), strengthening the hypothesis of negative regulation by miR-21-5p on FDX1.

Taken together, all these data suggest that miR-21-5p acts as the upstream regulator of FDX1 to drive ccRCC development.

### 3.7. Immune Cell Infiltration and Prognostic Prediction Model under the miR-21-5p/FDX1 Axis

Next, we sought to investigate the infiltration of immune cells under the miR-21-5p/FDX1 axis in TCGA-KIRC. Overall, T cells and macrophages, as the major populations of immune cells, were comparably infiltrated between the miR-21-5p_high and miR-21-5p_low masses (Figure 7A,B). Among the T cells, CD4^+^ T memory resting cells were less infiltrated in the miR-21-5p_high cohort (Figure 7C). On the contrary, tumor-promoting immune cells, M2-type macrophages, and T regulatory cells were abundantly enriched in miR-21-5p_high samples (Figure 7C). In addition, we also found that miR-21-5p expression was positively correlated with the expression levels of immune checkpoints such as PDCD1 (PD-1) and CTLA4, while FDX1 displayed a negative correlation with these genes (Figure 7D), indicating that the miR-21-5p/FDX1 axis may have the potential to affect the efficacy of immune therapies.

## 4. Discussion

Recently, the prognostic values of the cuproptosis signature and its associated tumor microenvironment have been recognized [20,21,22,23,24,25,26,27,28,29]. In addition, several studies have highlighted the tumor-suppressing role of FDX1 in ccRCC development [20,29,30]. However, how cuproptosis is regulated in ccRCC is not fully understood. In this study, we found that cuproptosis-resistant patients had a shorter survival rate and suffered from worse clinicopathological features, including the T stage, grade, and metastatic and alive status, as compared to cuproptosis-sensitive controls. We also generated the miR-21-5p/FDX1 axis and experimentally verified the tumor-suppressive role of FDX1 in ccRCC cells. A tight correlation between miR-21-5p/FDX1 expression and the infiltration of immune cells, especially the CD4^+^ T cell population, was clearly observed in ccRCC. Overall, our study firstly models the miR-21-5p/FDX1 axis and validates its cuproptosis-dependent and -independent roles in ccRCC development, providing a rationale for developing targeted therapies against this disease.

Resistance to cell death is a key feature of cancer cells. For instance, cancer prefers to express a high level of Bcl-2 [31,32], a pro-survival molecule, as one mechanism to overcome mitochondria-mediated apoptotic death, which leads to the development of Bcl-2 inhibitors as one potential management strategy to treat cancers [33,34]. Studies have also demonstrated that the pyroptosis determinants DFNA5 and GSDME were lowly expressed in some tumor cells [35,36], making them less responsive to pyroptotic-induced cell death. Therefore, it is reasonable to screen small molecules that could upregulate the expression levels of DFNA5/GSDME to induce pyroptosis, for cancer treatments. Herein, our study indicated that the cuproptosis master regulator FDX1 was dramatically decreased in ccRCC tumors, which was further silenced as the tumors progressed to an advanced stage, suggesting that ccRCC tumors may poorly respond to copper-induced cell death and choose to inhibit physiological cuproptosis for better survival. Herein, we identified miR-21-5p as a regulator of FDX1, suggesting that the delivery of a miR-21-5p inhibitor would have therapeutic value. However, it remains a big challenge to successfully deliver a miRNA inhibitor in the human body. Alternatively, other appropriate drugs would have been taken into therapeutic consideration for ccRCC if they bore the capacity to recover FDX1 expression.

The expression levels of FDX1, DLD, DLAT, PDHB, and PDHA1 were evidently decreased in ccRCC tumors as compared to normal kidney tissues, suggesting an abnormal regulation of cuproptosis regulators in ccRCC carcinogenesis. Epigenetic regulation, including DNA methylation and histone modification, is considered one critical mechanism responsible for gene expression, facilitating cancer progress [37]. However, a contradictory conclusion was established based on the analysis of DNA methylation levels in the promoter areas of these genes, suggesting that other mechanisms instead of epigenetic regulation account for the differential expression pattern of these genes between ccRCC and normal tissue. Indeed, we identified miR-21-5p as one upstream factor regulating FDX1 expression. However, whether miRNAs are also involved in the regulation of other cuproptosis regulators awaits further exploration. In addition to miR-21-5p-mediated gene silencing, other mechanisms responsible for the FDX1 reduction and regulation of other cuproptosis regulators in ccRCC should be investigated. For example, noncoding RNA-mediated mRNA alterations have been widely studied and are tightly involved in cancer development [38]. Recent studies have highlighted RNA modifications or epitranscriptomic modifications such as m^6^A (N6-methyladenosine) and m^1^A (N1-methyladenosine), which influence mRNA levels by altering their stability or translation [39,40].

Experimental evidence illustrated that knockdown of FDX1 significantly increased the cell invasion of OSRC-2 and ACHN cells, suggesting that FDX1 has a cuproptosis-independent function. Indeed, GSEA showed that the JAK_STAT pathway was enriched in ccRCC patients with low FDX1 expression, suggesting that FDX1 may have the potential to influence this well-known oncogenic pathway [41]. Indeed, our data clearly elucidated that knockdown of FDX1 significantly increased the phosphorylation levels of STAT3 at Y705, indicating that STAT3 signaling may be involved in the FDX1-mediated cuproptosis-independent phenotypic changes in ccRCC cells. The conclusion of the cuproptosis-independent role of FDX1 is consistent with previous results from other researchers, which showed that knockdown of FDX1 failed to alter cell growth and cell apoptosis in lung cancer cells but reduced ATP production [42]. Since cuproptosis is a death-related process, we therefore hypothesize that this phenomenon may have two possible explanations: first, the physiological level of cuproptosis is very low so that manipulation of FDX1 has negligible effect on this type of cell death; second, FDX2, another Fe-S cluster protein, may replace FDX1 to play a major role in regulating cuproptosis in lung cancer cells.

To sum up, we analyzed the potential role of physiological cuproptosis in ccRCC initiation and progression and generated cuproptosis risk scores for the overall survival prediction of this disease. Additionally, we identified the miR-21-5p/FDX1 axis in ccRCC and evaluated its potential contribution to the ccRCC microenvironment. Our study highlights that targeting miR-21-5p/FDX1 may be a good therapeutic option for ccRCC treatment.

## Figures and Tables

**Figure 1 cells-12-00173-f001:**
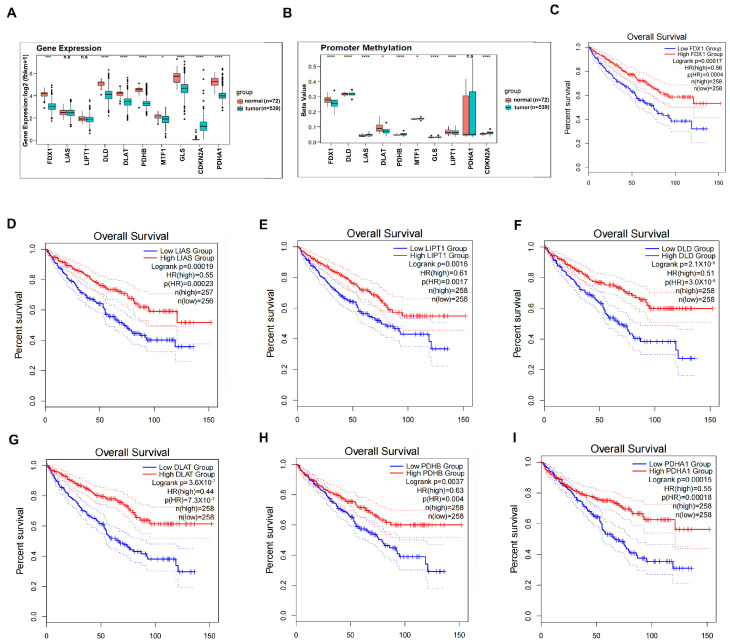
Clinical significance of cuproptosis regulators in ccRCC. (**A**) Expression levels of cuproptosis regulators between ccRCC and normal kidney tissues. (**B**) Methylation on promoter regions of cuproptosis regulators between ccRCC and normal kidney tissues. (**C**–**I**) High expression of FDX1 (**C**), DLAT (**D**), DLD (**E**), PDHB (**F**), LIAS (**G**), LIPT1 (**H**), and PDHA1 (**I**) predicted longer survival in ccRCC according to Kaplan–Meier survival analyses. * *p* < 0.05, *** *p* < 0.001, **** *p* < 0.0001; n.s. = no significance.

**Figure 2 cells-12-00173-f002:**
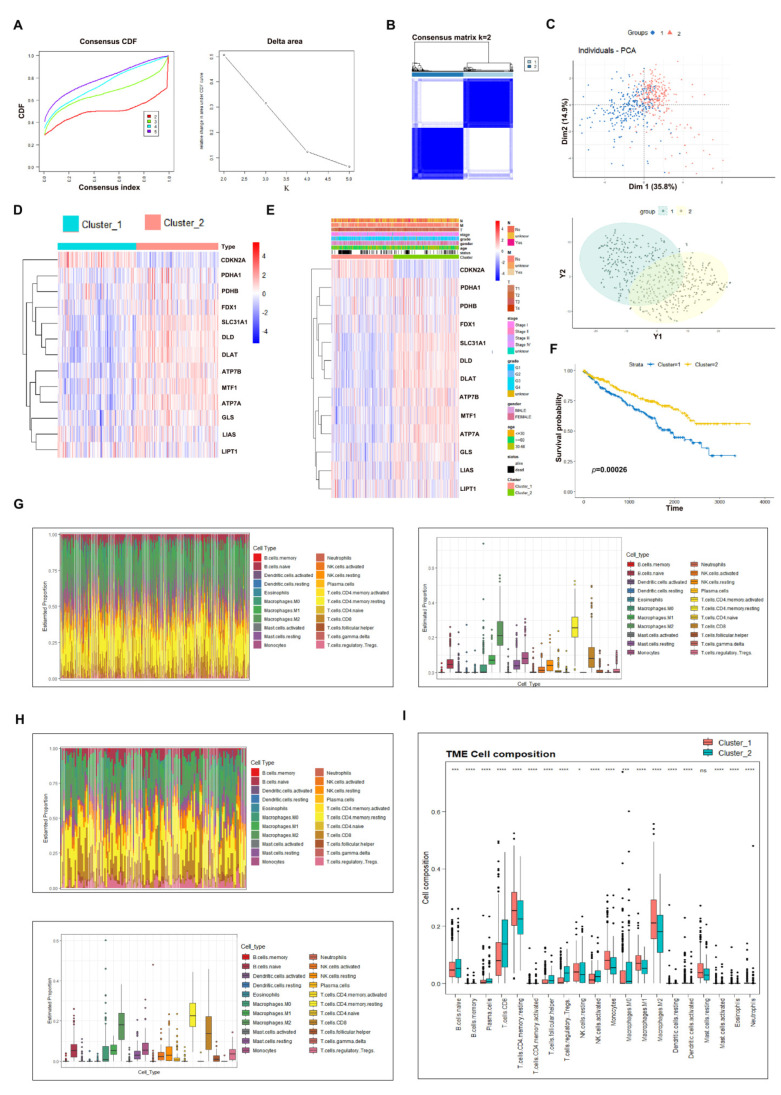
Consensus clustering analysis to evaluate the potential role of cuproptosis in ccRCC. (**A**) CDF and delta area analysis of category number *k*. (**B**) Consensus matrix when *k* = 2. (**C**) PCA analysis of CSS and CRS. (**D**) Expression heatmap of cuproptosis regulators in CSS and CRS. (**E**) A comparison of clinicopathological features between CSS and CRS. (**F**) CSS patients experienced longer survival as compared to CRS counterparts. (**G**) Estimated proportion of immune cells in CSS patients. (**H**) Estimated proportion of immune cells in CRS patients. (**I**) Comparison of immune cell infiltration between CSS and CRS patients. * *p* < 0.05, *** *p* < 0.001, **** *p* < 0.0001; n.s. = no significance.

**Figure 3 cells-12-00173-f003:**
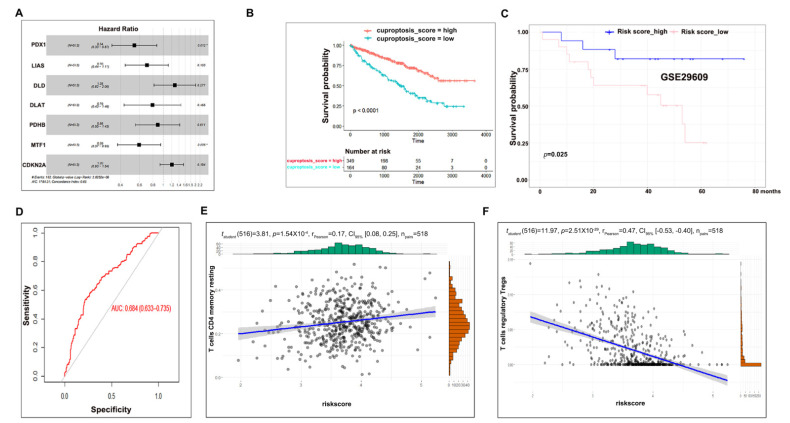
Cuproptosis risk score has a prognostic value in ccRCC. (**A**) Multivariate Cox regression analysis of seven cuproptosis regulators. (**B**) Cuproptosis_score high patients harbored longer survival as compared to cuproptosis_score low controls. (**C**) A fitness test of cuproptosis risk score in another independent dataset. (**D**) AUC analysis to evaluate the analysis of C. (**E**) Correlation between the CD^4+^ T memory resting cell population and the cuproptosis risk score. (**F**) Correlation between the T regulatory Treg cell population and the cuproptosis risk score.

**Figure 4 cells-12-00173-f004:**
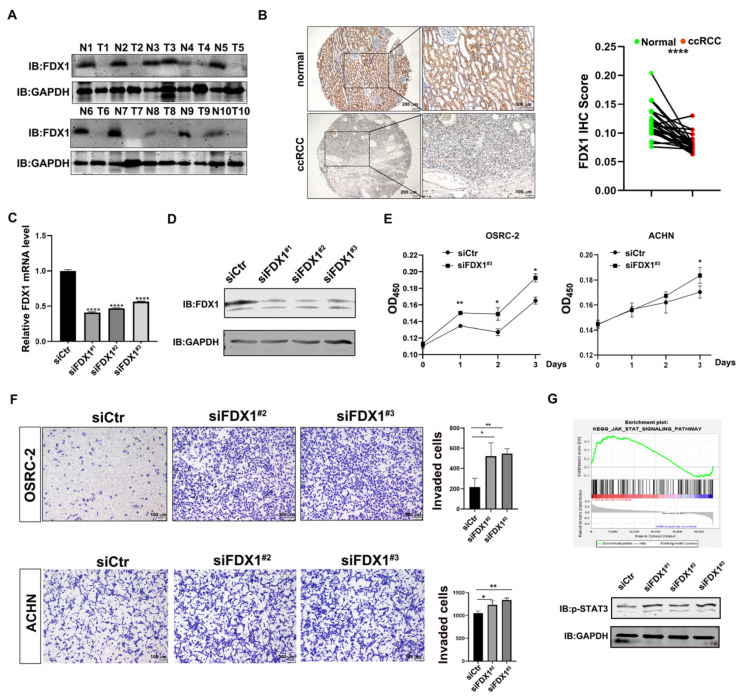
Experimental validation supports the tumor-suppressive role of FDX1 in ccRCC. (**A**) Western blotting to examine FDX1 expression in 10 paired ccRCC samples. (**B**) IHC staining of FDX1 expression in ccRCC tissue microarray (27/62 ccRCC with the paired adjacent tissues). Left, representative image of IHC staining. Right, statistical analysis of IHC staining. Scale bar: 200 μm for the left images and 100 μm for the right images. (**C**,**D**) Knockdown efficiency of FDX1 in OSRC-2 cells via qPCR (**C**) and WB (**D**). (**E**) Knockdown of FDX1 significantly increased cell growth of OSRC-2 and ACHN cells. (**F**) Knockdown of FDX1 remarkably increased cell invasion of OSRC-2 and ACHN cells. Left, representative image of invaded cells. Right, statistical analysis of invaded cells. Scale bar = 100 μm. (**G**) Top, GSEA analysis showed that JAK_STAT signaling was enriched in FDX1_low ccRCC patients. Bottom, FDX1 siRNA activated STAT3 signaling in OSRC-2 cells. GAPDH served as a loading control. * *p* < 0.05, ** *p* < 0.01, **** *p* < 0.0001.

**Figure 5 cells-12-00173-f005:**
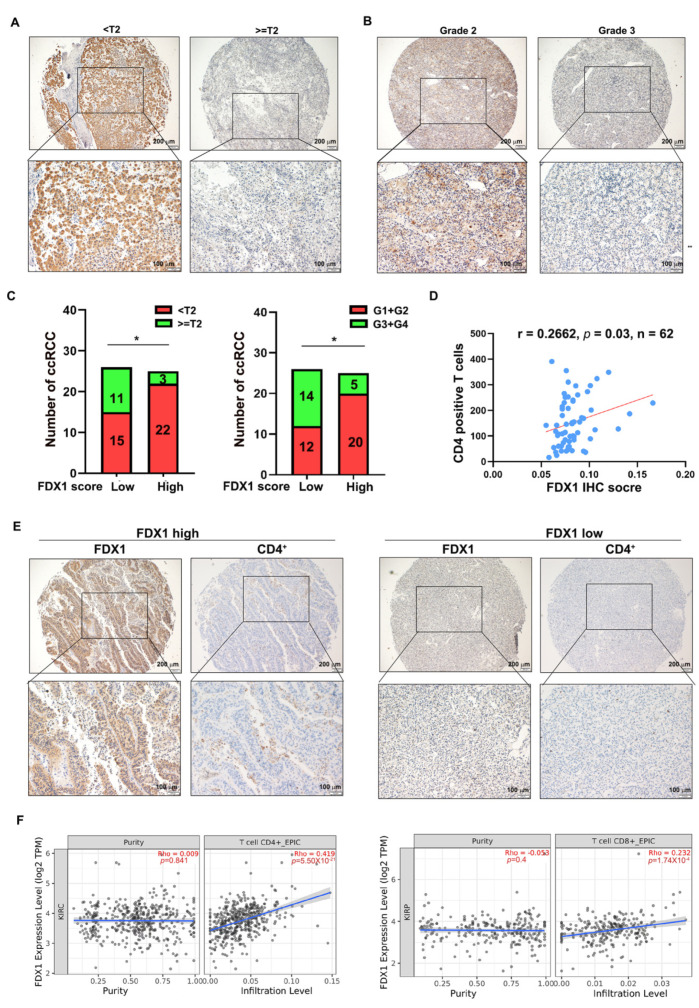
FDX1 correlates with the infiltration of CD4^+^ T cells in ccRCC. (**A**) Representative IHC images of FDX1 in different histological T stages of ccRCC. Scale bar: 200 μm for the top images and 100 μm for the bottom images. (**B**) Representative IHC images of FDX1 in different grades of ccRCC. Scale bar: 200 μm for the top images and 100 μm for the bottom images. (**C**) FDX1 was expressed at a higher level in ccRCC patients with histological T stage < T2 (left) or grade G1 + G2 (right) compared to their corresponding counterparts. *n* = 25 in T stage < T2; *n* = 26 in stage ≥ T2; *n* = 25 in G1 + G2; *n* = 26 in G3 + G4. (**D**) The FDX1 IHC score was positively correlated with the infiltration of CD4^+^ T cells in ccRCC (r = 0.2662, *p* = 0.03, *n* = 62). (**E**) Representative images of the correlation between FDX1 intensity and CD4^+^ T cell infiltration. Scale bar: 200 μm for the top images and 100 μm for the bottom images. (**F**) Immune cell infiltration analysis of the TCGA-KIRC dataset using *TIMER 2.0* showed that FDX1 expression was positively correlated with CD4^+^ and CD8^+^ T cell populations. * *p* < 0.05.

**Figure 6 cells-12-00173-f006:**
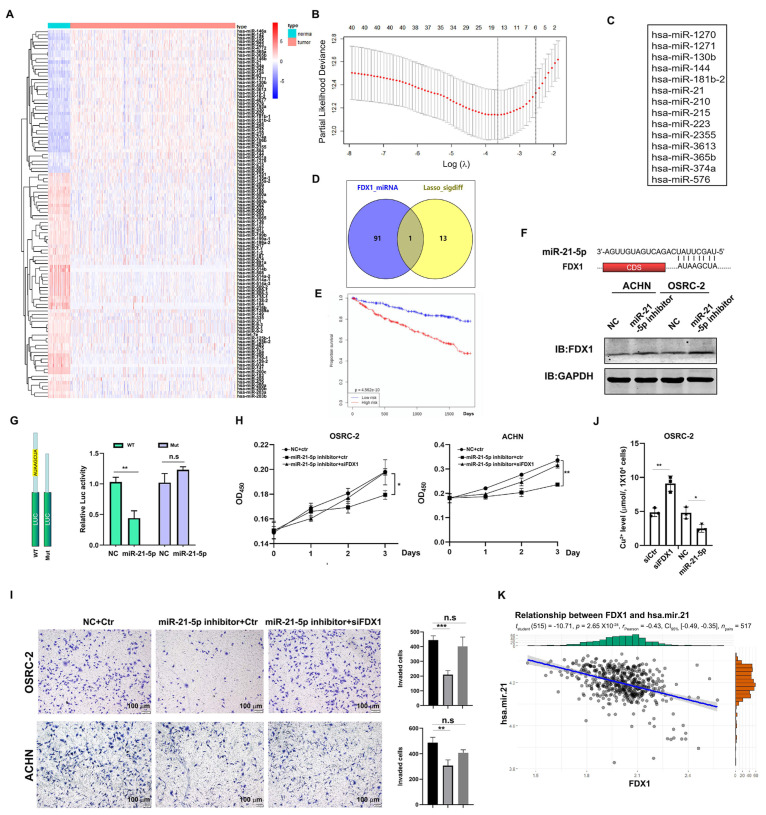
Identification of miR-21-5p as the upstream regulator of FDX1. (**A**) A heap map of miRNAs differentially expressed between normal kidney tissues and ccRCC. (**B**) Prognosis-based LASSO analysis of the differential miRNAs. (**C**) A list of 14 miRNAs highly related to the prognosis. (**D**) A Venn diagram of 14 miRNAs and FDX1-targeted miRNAs. (**E**) High expression of miR-21-5p was associated with a short overall survival of ccRCC. (**F**) Top, the predicted binding site between the 3′-UTR of FDX1 and miR-21-5p. Bottom, miR-21-5p inhibitor treatment elevated FDX1 expression in both ACHN and OSRC-2 cells. GAPDH was the loading control. (**G**) Luciferase report assay showed that miR-21-5p mimics remarkably suppressed the activity of 3′-UTR of FDX1. (**H**) miR-21-5p inhibitor-suppressed cell growth of ACHN and OSRC-2 cells could be rescued by FDX1 knockdown. (**I**) miR-21-5p inhibitor-suppressed cell invasion of OSRC-2 cells could be blocked by FDX1 knockdown. Left, representative invading cells. Right, statistical analysis. Scale bar = 100 μm. (**J**) Cu^2+^ level in OSRC-2 cells before and after miR-21-5p inhibitor or siFDX1 treatment. (**K**) miR-21-5p expression was inversely correlated with FDX1 in TCGA-KIRC. * *p* < 0.05, ** *p* < 0.01, *** *p* < 0.001; n.s. = no significance.

**Figure 7 cells-12-00173-f007:**
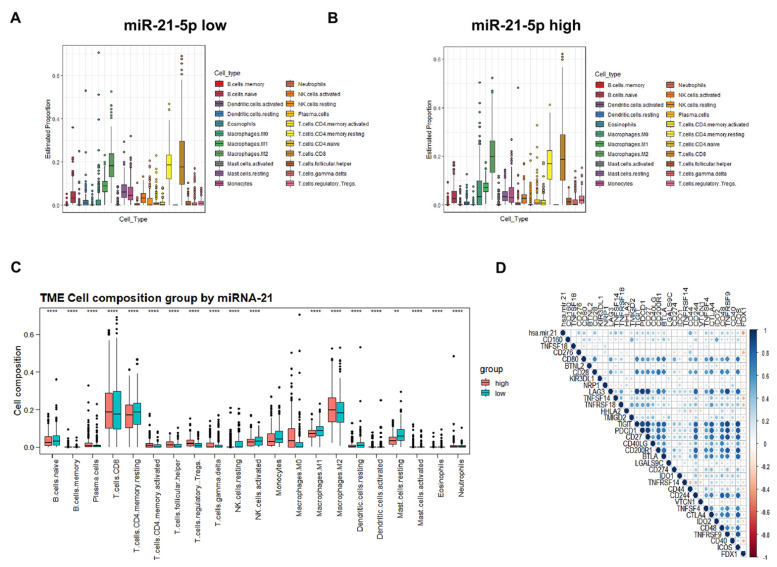
Immune cell infiltration and prognostic prediction model under the miR-21-5p/FDX1 axis. (**A**) Estimated proportion of tumor microenvironmental components in miR-21-5p_low ccRCC patients. (**B**) Estimated proportion of tumor microenvironmental components in miR-21-5p_high ccRCC. (**C**) Comparison of tumor microenvironmental components between miR-21-5p_high and miR-21-5p_low ccRCC. (**D**) Correlation of miR-21-5p/FDX1 with immune checkpoints. ***p* < 0.01; *****p* < 0.0001.

## Data Availability

The raw data will be provided by the corresponding author upon reasonable request.

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
