# Peer review of "Cuproptosis-Related MiR-21-5p/FDX1 Axis in Clear Cell Renal Cell Carcinoma and Its Potential Impact on Tumor Microenvironment"

_cells, 2022, doi:10.3390/cells12010173_

Round 1

Reviewer 1 Report

The manuscript by Xie, Cheng and Yu et al. “Cuproptosis related MiR-21-5p/FDX1 axis construction in clear cell renal cell carcinoma and its potential impact on tumor micro-environment” combines bioinformatic and experimental analyses of the cuproptosis, a novel cell death pathway in renal carcinoma. It is an extension of the recent findings of cuproptosis beyond blood tumor cell lines. The experiments were carried out with sufficient expertise to support their general conclusions, worthy of timely publication in this journal. However, technical English writing requires substantial improvement. There is no description of how cellular copper level was determined (Fig. 6J). Other examples include but not limited to the following. The word “construction” in the title as well as other places does not make sense, a simple statement of signaling axis in renal cancer should suffice. Line 55, “approval” is not appropriate, as the findings should have led to identification of TKIs to treat patients. Line 113, there is no such a company in China called “anti-Hela biotechnology company”. Line 122, first letter needs to be capitalized. Line 124, misspelling of leupeptin and aprotinin. Line 193, there is a grammatical error. Line 277, “patients” should be “cells” or “masses”, similarly in line 313. This extends to line 383, should be “enriched in miR-21-5p_high samples”. In discussion, line 450-453 regarding the cuproptosis-in-dependent role of FDX1 in lung cancer cells requires further deliberation as there is FDX2 which might play a role or both genes are needed for cuproptosis in lung cancer cells.

Author Response

Reviewer 1#

Q1. The manuscript by Xie, Cheng and Yu et al. “Cuproptosis related MiR-21-5p/FDX1 axis construction in clear cell renal cell carcinoma and its potential impact on tumor micro-environment” combines bioinformatic and experimental analyses of the cuproptosis, a novel cell death pathway in renal carcinoma. It is an extension of the recent findings of cuproptosis beyond blood tumor cell lines. The experiments were carried out with sufficient expertise to support their general conclusions, worthy of timely publication in this journal.

Ans: We appreciate your positive comments. Thanks

Q2. However, technical English writing requires substantial improvement.

Ans: Thanks for the suggestion. We already sent the manuscript out for a language revision provided by MDPI. The proof of this service as attached in the supplementary files. Thanks

Q3. There is no description of how cellular copper level was determined (Fig. 6J).

Ans: We thank your careful review. The description of how cellular copper level was measured was supplemented in the Materials and Methods section (page 4, line 175-179). Thanks

Q4. Other examples include but not limited to the following. The word “construction” in the title as well as other places does not make sense, a simple statement of signaling axis in renal cancer should suffice. Line 55, “approval” is not appropriate, as the findings should have led to identification of TKIs to treat patients. Line 113, there is no such a company in China called “anti-Hela biotechnology company”. Line 122, first letter needs to be capitalized. Line 124, misspelling of leupeptin and aprotinin. Line 193, there is a grammatical error. Line 277, “patients” should be “cells” or “masses”, similarly in line 313. This extends to line 383, should be “enriched in miR-21-5p_high samples”.

Ans: Again, we appreciate your careful review. As your concerns, we already made the appropriate corrections. OSRC-2 (Cat: IM-H061) and ACHN (Cat: IM-H058), which represents VHL null and VHL wild-type RCC cell line respectively, were purchased from Immocell Biotechnology company (www.immocell.com, Xiamen,  China).In addition, our manuscript has received language service provided by MDPI. Now we believe this revised manuscript is language qualified for publication. Thanks.

Q5. In discussion, line 450-453 regarding the cuproptosis-in-dependent role of FDX1 in lung cancer cells requires further deliberation as there is FDX2 which might play a role or both genes are needed for cuproptosis in lung cancer cells.

Ans: Thanks for your comment. The discussed literature demonstrates that depletion of FDX1 has no effect on the cell growth of lung cancer cells. Since cuproptosis is a death-related process, we therefore hypothesize that this phenomenon has two possible explanations: first, the physiological level of cuproptosis is very low so that manipulation of FDX1 has negligible effect on this type of cell death; second, FDX2, another Fe-S cluster protein, may replace FDX1 to play a major role in regulating cuproptosis in lung cancer cells. You can reach this deliberated discussion in page 18 line 508-513. Thanks.

Reviewer 2 Report

In the manuscript "Cuproptosis related MiR-21-5p/FDX1 axis construction in clear two cell renal cell carcinoma and its potential impact on tumour micro-environment", Mingyue Xie and colleagues analyze the function of the FDX1 protein - cuproptosis master regulator – in the ccRCC tumour initiation and progression. The authors have evaluated the association of the MiR-21-5p/FDX1 axis with immune cell infiltration.

This study is an extensive research work with a clear objective and planned methodology, combining the bioinformatic methods with the cell lines experiments and analysis of the tissue samples. 

However, I have a few concerns. 

General remarks  

First, the introduction regarding the cuproptosis is too brief. Could You provide a more detailed description of the cuproptosis and the physiological function of the FDX1 gene?  

Could you also elaborate on how far the process of cuproptosis has been researched in different cancers?  

In the manuscript, in many sites, the experiments with human tissues are mentioned (sections 2.10? 2.13, 3.4, 3.5, 3.7?); unfortunately, there is no characteristic of the patient cohort involved in the study. Also, the patient consent or ethical committee approval information is missing. Please provide the number of the ethical committee approval and the table with the characteristics of the patients involved in the study (gender, age, tumour size, cancer staging).  

While reading the manuscript, I had problems identifying the type of analysis done in each study section. Please indicate more clearly what results come from bioinformatic analysis of data sets, what has been researched using cell lines, and where the patient's tissue was analyzed. It is also essential to include information on the number of patient samples that were analyzed in individual experiments because different amounts appear in the text (section 3.4 - 10 pairs of ccRCC, Figure 4B – 27 pairs, section 3.5, and Figure 5 – 51 tissues?; 3.6?; 3.7) 

In the Material and Method section: 

In the reviewer's opinion, the paragraphs related to the bioinformatic analysis (2.3-2.6) should be grouped in order not to confuse the methods described in the experimental part. Also, it is essential to include the date of accession for the data sets downloaded from TCGA, GEO, and KIRC.  

In the section related to the calculation of the cuproptosis risk score, it would be good to elaborate on why the following genes: FDX1, LIAS, DLD, DLAT, PDHB, MTF1 and CDKN2A were taken into consideration. Or consider including a detailed description of those genes in the introduction. 

In section 2.5, please add the information on which regulators will be analyzed and visualized. 

Regarding the experiments with cell lines, could You provide short characteristics of the selected cancer cell lines and include the cat numbers of the purchased cell lines? Is the "humin hood" alright?  

In section 2.8. please include the siRNA sequences.  
Section 2.12 - please provide the manufacturer's name for the J software. 

Section 2.13 - how many sections were analyzed, and from how many patients?  

Results 

Section 3.1 - how has it been demonstrated that 5 out of 7 genes tested are downregulated? In Fig 1.A, due to the lack of unit on the Y axis, all the analyzed genes seemed to be upregulated, not downregulated. Was the gene expression normalized to a specific gene?  

In the sentence "Significantly, high expression levels of these seven positive regulators of cuproptosis served as good indicators of the overall survival in ccRCC patients based on Kaplan–Meier survival analyses (Fig. 1C-I), indicating resistance to cuproptosis is one potential mechanism driving ccRCC progression" are the Authors referring to the increased expression in cancer tissues vs non-cancerous tissue? Or vs a reference gene?  

Figure 1.A and 1.B. provide the unit description on the Y axis. 

Figure 1(A,B), Figure2,3 and 4 – please provide bigger fonts because the captions are hard to read. 

Section 3.2 - the conducted immune cells infiltration analysis was performed using which data set? How was the proportion of immune cell estimation performed in CSS/CRS patients? 

Section 3.3 - please elaborate more on the definition of the classification of the ccRCC patients into cuproptosis_score high and low groups. Which data set was used to build the cuproptosis risk score? 

Section 3.4 - please provide more information on the ccRCC patients' histopathology. Were the 27 samples (Fig.4B) obtained from 10 patients, or is this a separate cohort?

What was the characteristic of the siRNA used to knock down the RDX1 gene? 

Section 3.5 - How was the FDX1 expression (immunodepression) level assessed in the samples, and how was the low/high FDX1 score calculated? In Figure 5.3, if there is a proportion of samples (%), please change the Y-axis range.  

Editorial comments: 

Please rebuild the following sentences: 

"This deregulation leads to the approval of tyrosine kinase inhibitors such as sunitinib and pazopanib to treat ccRCC patients, which displays clinical benefits". 

"Further investigation of the DNA methylation levels on the promoter regions of these genes demonstrated that the alterations of these genes were not simply caused by epigenetic regulation (Fig. 1B) and post-transcriptional regulations were involved." - you have analyzed only methylation. Further, the manuscript analyses the miR-21-5p effect on the FDX1 expression, so You are pointing out the epigenetic regulation. " 

“In agreement with the tumour suppressive role of FDX1, miR-21-5p inhibitor profoundly slowed down cell growth and cell invasion of ACHN and OSRC-2 cells, which could be alleviated by the reintroduction of FDX1 siRNA (Fig. 6H, ").” - the fragment hard to follow, please rewrite the sentence to make it clear. 

Author Response

Reviewer 2#

Q1. In the manuscript "Cuproptosis related MiR-21-5p/FDX1 axis construction in clear two cell renal cell carcinoma and its potential impact on tumour micro-environment", Mingyue Xie and colleagues analyze the function of the FDX1 protein - cuproptosis master regulator – in the ccRCC tumour initiation and progression. The authors have evaluated the association of the MiR-21-5p/FDX1 axis with immune cell infiltration.This study is an extensive research work with a clear objective and planned methodology, combining the bioinformatic methods with the cell lines experiments and analysis of the tissue samples. 

Ans: We appreciate your positive comments. Thanks

Q2. However, I have a few concerns. 

General remarks  

First, the introduction regarding the cuproptosis is too brief. Could You provide a more detailed description of the cuproptosis and the physiological function of the FDX1 gene?  Could you also elaborate on how far the process of cuproptosis has been researched in different cancers?  

Ans: We highly appreciate your comments. A more detailed introduction regarding cuproptosis and FDX1 was provided in the revised manuscript. Thanks.

Q3. In the manuscript, in many sites, the experiments with human tissues are mentioned (sections 2.10? 2.13, 3.4, 3.5, 3.7?); unfortunately, there is no characteristic of the patient cohort involved in the study. Also, the patient consent or ethical committee approval information is missing. Please provide the number of the ethical committee approval and the table with the characteristics of the patients involved in the study (gender, age, tumour size, cancer staging).  

Ans: Your careful review is highly appreciated. The ethical committee approval of this study was provided in the supplementary files. Also, the clinicopathological characteristics of ccRCC patients in tissue micro-array was listed in Table 2. Briefly, fresh samples (Patient ID:1-10) were utilized for western blotting in Figure 4A. Paraffinized tumor samples with paired adjacent normal tissues (Patient ID 1-27) were used to perform statistical analysis of FDX1 IHC score in Figure 4B. And paraffinized tumor samples (Patient ID:1-51) with clear histological T stage and grade confirmed by pathologist were used to analyze the correlation between FDX1 expression and histological T stage (Figure 5C, left) or Grade (Figure 5C, right). The detailed description can be reached in the revised manuscript. Thanks.

Q4. While reading the manuscript, I had problems identifying the type of analysis done in each study section. Please indicate more clearly what results come from bioinformatic analysis of data sets, what has been researched using cell lines, and where the patient's tissue was analyzed. It is also essential to include information on the number of patient samples that were analyzed in individual experiments because different amounts appear in the text (section 3.4 - 10 pairs of ccRCC, Figure 4B – 27 pairs, section 3.5, and Figure 5 – 51 tissues?; 3.6?; 3.7) 

Ans: Thanks for the careful review. We already made the description of individual analysis more detailed in the revised manuscript. Thanks.

Q5. In the Material and Method section: 

In the reviewer's opinion, the paragraphs related to the bioinformatic analysis (2.3-2.6) should be grouped in order not to confuse the methods described in the experimental part. Also, it is essential to include the date of accession for the data sets downloaded from TCGA, GEO, and KIRC.  

Ans: Thanks for the careful review. As requested, all bioinformatic analyses were grouped and the related datasets or sources were provided in the revised manuscript. Thanks.

Q6. In the section related to the calculation of the cuproptosis risk score, it would be good to elaborate on why the following genes: FDX1, LIAS, DLD, DLAT, PDHB, MTF1 and CDKN2A were taken into consideration. Or consider including a detailed description of those genes in the introduction. 

Ans: Thanks for the careful review. The rationale to consider these seven genes was described in page 9 line 290-298. Briefly, univariate Cox regression analysis towards ten cuproptosis regulators in TCGA-KIRC ccRCC patients was performed and seven members (FDX1, LIAS, DLD, DLAT, PDHB, MTF1 and CDKN2A) were further enrolled to multivariate Cox regression analysis owing to their significant correlations with the overall survival of ccRCC patients. As a result, a cuproptosis risk score was generated for each individual ccRCC patient by weighting the prognostic input of individual genes based on their Cox regression coefficients and expression levels. Finally, the cuproptosis risk score was calculated using the following equation: risk score= ∑ gene expression X Cox regression coefficient. Thanks.

Q7. In section 2.5, please add the information on which regulators will be analyzed and visualized. 

Ans: We appreciate your careful review. 10 regulators including FDX1, DLD, DLAT, PDHB, LIAS, LIPT1, MTF1, GLS, PDHA1 and CDKN2A were added in this section. Thanks.   

Q8. Regarding the experiments with cell lines, could You provide short characteristics of the selected cancer cell lines and include the cat numbers of the purchased cell lines? Is the "humin hood" alright?  

Ans: Your comment is highly appreciated. OSRC-2 (Cat: IM-H061) and ACHN (Cat: IM-H058), representing VHL null and VHL wild-type RCC cell line respectively, were purchased from Immocell Biotechnology company (Xiamen, China). We are sorry for the typo error: “humind” should be “humidified”. We already made these corrections in the revised manuscript. Thanks

Q9. In section 2.8. please include the siRNA sequences.  

Ans: As request, the sequences of three siRNAs against FDX1 were provided in the revised Methods section. Thanks

Q10. Section 2.12 - please provide the manufacturer's name for the J software. 

Ans: Thanks for the careful review. Image J software is downloaded from National Institutes of Health (NIH), which was provided in the revised manuscript. Thanks

Q11. Section 2.13 - how many sections were analyzed, and from how many patients?  

Ans: We appreciate your careful review. Our micro-array contains 62 ccRCC and 27 adjacent kidney tissues. For the correlation between FDX1 and CD4 positive T cell infiltration, 62 ccRCC sections were used for the analysis. For the correlation between FDX1 and histological T stage or grade, 51 ccRCC sections were enrolled because the T stage and grade of other 11 ccRCC are not confidently confirmed by pathologist. The detailed information was provided in the revised manuscript. Thanks.

Q12. Results 

Section 3.1 - how has it been demonstrated that 5 out of 7 genes tested are downregulated? In Fig 1.A, due to the lack of unit on the Y axis, all the analyzed genes seemed to be upregulated, not downregulated. Was the gene expression normalized to a specific gene?  

Ans: We are sorry for this confusing description. We already corrected the inappropriate statements as “the expression levels of 5/7 positive regulators of cuproptosis (FDX1, DLD, DLAT, PDHB and PDHA1) were robustly decreased in ccRCC as compared to normal kidney controls, whereas the negative regulator CDKN2A was profoundly increased in ccRCC patients as compared to the adjacent non-cancerous tissues” and added the unit on the Y axis to Figure 1A. Thanks.

Q13. In the sentence "Significantly, high expression levels of these seven positive regulators of cuproptosis served as good indicators of the overall survival in ccRCC patients based on Kaplan–Meier survival analyses (Fig. 1C-I), indicating resistance to cuproptosis is one potential mechanism driving ccRCC progression" are the Authors referring to the increased expression in cancer tissues vs non-cancerous tissue? Or vs a reference gene?

Ans: We appreciate your comment. TCGA-KIRC ccRCC patients were classified into high_group and low_group using the median expression level of one certain gene as a cutoff and overall survival was compared between these two groups. Thanks.  

Q14. Figure 1.A and 1.B. provide the unit description on the Y axis. 

Figure 1(A,B), Figure 2,3 and 4 – please provide bigger fonts because the captions are hard to read. 

Ans: We are sorry for this confusing description and the small font size. The unit of Y axis in Figure 1A is log2(fpkm+1), which was already provided in the revised Figs. The Beta value in Figure 1B, which ranges from 0 (unmethylated) to 1 (fully methylated), is already the unit of DNA methylation level. In addition, bigger fonts were re-provided in Fig 1(A,B) and Fig 2,3 to make it easy to read. Thanks.

Q15. Section 3.2 - the conducted immune cells infiltration analysis was performed using which data set? How was the proportion of immune cell estimation performed in CSS/CRS patients? 

Ans: We appreciate your comment. Immune cell infiltration analysis was conducted on CSS and CRS, classified by consensus clustering. Briefly, the LM22 signature matrix and CIBERSORT R script were downloaded from this literature [Robust enumeration of cell subsets from tissue expression profiles. Nat Methods. 2015 May;12(5):453-7], and the samples from CSS and CRS were used to evaluate the level of immune cell infiltration based on the signature matrix related to the specific population of immune cells and the CIBERSORT algorithm. The detailed information can be reached in page 3 line 125-129. Thanks.

Q16. Section 3.3 - please elaborate more on the definition of the classification of the ccRCC patients into cuproptosis_score high and low groups. Which dataset was used to build the cuproptosis risk score? 

Ans: Your careful review is highly appreciated. TCGA-KIRC was used to build the cuproptosis risk score and an independent ccRCC dataset GEO29609 was used to test the fitness of the cuproptosis risk score. And a risk score = 3.52 as a cutoff point based on the analysis from survcutpoint in survminer R package. ccRCC with risk score>3.52 was classified into cuproptosis_score high group while ccRCC with risk score<=3.52 was classified into cuproptosis_score high group. Thanks.

Q17. Section 3.4 - please provide more information on the ccRCC patients' histopathology. Were the 27 samples (Fig.4B) obtained from 10 patients, or is this a separate cohort?

Ans: We appreciate your comment. The detailed information of ccRCC patients used in this study was provided as Table 2 in the revised manuscript. 10 ccRCC samples with paired kidney tissues used in Figure 4A were fresh ones of 10/27 samples in Figure 4B. The details were described in Ans to Q3. Thanks.

Q18. What was the characteristic of the siRNA used to knock down the FDX1 gene? 

Ans: The three siRNAs used to knock down FDX1, including sequences and targeting sites, were provided in the Materials and Methods section. Thanks

Q19. Section 3.5 - How was the FDX1 expression (immunodepression) level assessed in the samples, and how was the low/high FDX1 score calculated? In Figure 5.3, if there is a proportion of samples (%), please change the Y-axis range.  

Ans: The brown color (DAB positive) in section was isolated by IHC tool-box plugin  and calculated by Image J to get an average intensity of the DAB positive pixels, which was utilized as IHC score to represent gene expression. The median level of FDX1 IHC score was set as the cutoff to classify ccRCC tumor sections into FDX1_high and FDX1_low groups. Thanks

Round 2

Reviewer 2 Report

Dear Authors,

Thank You for including additional explanations in the manuscript.

The amended version of the manuscript is much more understandable, mainly when presenting the results of the bioinformatic analysis. 

There is essential information missing - the ethical committee approval. In the supplementary materials, there is a document in Chinese, maybe this is the missing approval. Still, the information on the support is needed - the name of the institution issuing the approval and its number.

Could You please provide Figures 2, 3, 6 and 7 in a higher resolution, as the chart descriptions and legends are hard to read?

In the supplementary materials:
from which dataset the data for Table 1 were taken?
Please take a look at table 2 - there is "glade" - probably it should be "grade".

Author Response

Q1. There is essential information missing - the ethical committee approval. In the supplementary materials, there is a document in Chinese, maybe this is the missing approval. Still, the information on the support is needed - the name of the institution issuing the approval and its number.

Ans: Your careful review is highly appreciated. The information of this approval can be reached in Page 4 line 190-192. Thanks

Q2. Could You please provide Figures 2, 3, 6 and 7 in a higher resolution, as the chart descriptions and legends are hard to read?

Ans: We appreciate your comment. The images of Fig 2,3,6,7 with higher resolution were provided in the revised manuscript. Thanks.

Q3. In the supplementary materials:from which dataset the data for Table 1 were taken?
Please take a look at table 2 - there is "glade" - probably it should be "grade".

Ans: Thanks for your careful review. Table 1 was the analysis of TCGA-KIRC (Page 5 line 221 and the title of Table 1) and the typo error “glade” was corrected to “grade”. Thanks.